# The Safety and Efficacy of Sodium-Glucose Cotransporter-2 Inhibitors for Patients with Sarcopenia or Frailty: Double Edged Sword?

**DOI:** 10.3390/jpm14020141

**Published:** 2024-01-26

**Authors:** Ayami Naito, Yuji Nagatomo, Akane Kawai, Midori Yukino-Iwashita, Ryota Nakazawa, Akira Taruoka, Asako Takefuji, Risako Yasuda, Takumi Toya, Yukinori Ikegami, Nobuyuki Masaki, Yasuo Ido, Takeshi Adachi

**Affiliations:** 1Department of Cardiology, National Defense Medical College, Tokorozawa 359-8513, Japan; 2Department of Intensive Care, National Defense Medical College, Tokorozawa 359-8513, Japan

**Keywords:** sodium-glucose cotransporter-2 (SGLT-2) inhibitors, heart failure, frailty, sarcopenia

## Abstract

Sodium-glucose cotransporter-2 inhibitors (SGLT-2is) show cardiovascular protective effects, regardless of the patient’s history of diabetes mellitus (DM). SGLT2is suppressed cardiovascular adverse events in patients with type 2 DM, and furthermore, SGLT-2is reduced the risk of worsening heart failure (HF) events or cardiovascular death in patients with HF. Along with these research findings, SGLT-2is are recommended for patients with HF in the latest guidelines. Despite these benefits, the concern surrounding the increasing risk of body weight loss and other adverse events has not yet been resolved, especially for patients with sarcopenia or frailty. The DAPA-HF and DELIVER trials consistently showed the efficacy and safety of SGLT-2i for HF patients with frailty. However, the Rockwood frailty index that derived from a cumulative deficit model was employed for frailty assessment in these trials, which might not be suitable for the evaluation of physical frailty or sarcopenia alone. There is no fixed consensus on which evaluation tool to use or its cutoff value for the diagnosis and assessment of frailty in HF patients, or which patients can receive SGLT-2i safely. In this review, we summarize the methodology of frailty assessment and discuss the efficacy and safety of SGLT-2i for HF patients with sarcopenia or frailty.

## 1. Introduction

Sodium-glucose cotransporter-2 inhibitors (SGLT-2is) are drugs that increase urinary sodium and glucose excretion by inhibiting the effect of SGLT-2 in the proximal renal tubules [1]. Accumulating evidence suggests that SGLT-2is show not only blood-glucose-lowering effects but also cardiovascular protective effects. The various mechanisms mediating its beneficial effect [2] include the diuretic effect by sodium discharge and osmotic diuresis [3,4], glomerular and tubular protection [5], increased erythropoiesis [3,6,7], sympathetic nervous system inhibition [8,9], improvement of myocardial energy metabolism [10,11], suppression of chronic inflammation [12] or oxidative stress [11], and weight reduction [13]. In large-scale randomized control trials such as EMPA-REG OUTCOME, CANVAS program, and DECLARE–TIMI 58, SGLT-2is, including empagliflozin, canagliflozin and dapagliflozin suppressed the composite outcome of cardiovascular death, nonfatal myocardial infarction, and stroke for the patients with type 2 diabetes mellitus (DM) and high risk of cardiovascular events (Table 1) [14,15,16]. As a result, the exploration of SGLT-2is’ beneficial effect was extended to the heart failure (HF) population. In the EMPEROR-Reduced and DAPA-HF trial [17,18], the risk of worsening HF events (hospitalization or urgent visit resulting in intravenous therapy for HF) or cardiovascular death were suppressed in the patients with heart failure and reduced ejection fraction (HFrEF) who received SGLT-2is compared to those who received a placebo. This beneficial effect of SGLT-2is on HFrEF patients has been observed regardless of the history of DM [18,19]. Furthermore, SGLT-2is successfully improved the clinical outcome even in patients with HF and preserved EF (HFpEF), although there had been no agents that demonstrated the prognostic benefit in this population until then [20,21]. In addition, SGLT-2is consistently provide evidence of HF event reduction in these studies, although the mortality benefit has been controversial [17,18,19,20,21]. Further, the treatment effect of SGLT-2is was not significantly influenced by EF [22]. Along with these research findings, SGLT-2is are recommended for patients with HF irrespective of EF in the AHA/ACC/HFSA guidelines [23] and ESC guidelines [24]. Despite these benefits, the concern surrounding the increasing risk of body weight loss, urogenital infection, hypoglycemia, volume depletion, bone fracture, and diabetic ketoacidosis has not yet been resolved [25]. Further, there have been significant concerns surrounding these adverse effects for elderly populations because of the increased susceptibility to side effects, impaired awareness of adverse events, poorer adherence and higher risk of falling. Among these adverse effects, weight loss and bone fracture might be derived from renal glucose excretion and energy loss by inhibiting SGLT-2. Thus, the safety of SGLT-2is in frail patients is still unclear.

The presence of frailty [27,28,29,30,31,32] or sarcopenia [33,34,35,36,37] is known as a prognostic aggravating factor in HF, leading to a higher risk of hospitalization and mortality. According to the remarkably accelerated aging of HF populations [38], the prevalence of frailty or sarcopenia has been dramatically increasing and is further expected to keep rising in the future [39]. Aging is the most significant contributing factor to frailty and these are deeply related to each other but not necessarily parallel. In addition, although there are scales widely used to assess frailty (Table 2), no scale has been established specifically for HF patients.

According to the American Diabetes Association Guideline recommendation [46], management of elderly DM patients requires individualized treatment targets that take account of their comorbidities because of the risk of hypoglycemia or ketosis resulting from the disruption of diet and medication. With the aforementioned risks, some trials indicate that SGLT-2is administration for elderly patients has similar or greater benefits for cardiovascular or renal function than younger patients [46]. Nevertheless, it is necessary to consider the characteristics of each racial group for worldwide consensus, especially Asian populations that have differences in body composition and cardiometabolic risk from Caucasian populations [47].

Hence, in this review, we summarize the methodology of frailty assessment and discuss the efficacy and safety of SGLT-2is for HF patients with frailty.

## 2. Definition and Etiology of Sarcopenia or Frailty

Sarcopenia and frailty are sometimes associated with a similar clinical picture but these two terms differ substantially in terms of their concept. Sarcopenia is a syndrome characterized by progressive and generalized loss of skeletal muscle mass and strength with a risk of adverse outcomes such as physical disability, poor quality of life and death [48,49]. According to the conceptual definition of sarcopenia by the European Working Group on Sarcopenia in Older People (EWGSOP), diagnosis of sarcopenia is made by the presence of both low muscle mass and low muscle function such as muscle strength or physical performance [50,51]. Further, it is defined as severe sarcopenia when all of these three components (low muscle mass, low strength and low physical performance) are present [50,51]. By the current recommendation, the assessment tool for sarcopenia is composed of muscle mass measured by Appendicular Skeletal Muscle Mass (ASM), muscle strength measured by grip or chair stand, and physical performance measured by gait speed, Short Physical Performance Battery (SPPB) [40], or Timed-Up and Go test (TUG) (Table 2) [41,51]. This recommendation focuses on European populations, while different diagnostic criteria have been proposed for Asian populations by the Asian Working Group for Sarcopenia (AWGS) [52], since body composition substantially differs between these ethnicities [47].

On the other hand, frailty is classically defined as the presence of three or more of the following criteria: unintentional weight loss (more than 4.5 kg in 1 year), slow gait speed, weak grip strength and self-reported physical exhaustion or measured low physical activity [45]. However, the concept of frailty has been broadened and is now defined as the deterioration of multidimensional and multisystem conditions characterized by decreased functional reserves and increased vulnerability to stress and acute adverse events [53]. Thus, it is a broad concept in contrast to sarcopenia, which focuses on muscle mass or weakness. Frailty includes a medical domain, a physical domain, a cognitive/depressive status domain, and a social domain [54]. Although there are various indices and scores proposed to quantify frailty which is the complex multisystem condition, there are two basic concepts of frailty, phenotype model and the cumulative deficit model [55]. The phenotype model is a measure of the presence of symptoms or physical functions such as activity of daily living (ADL), which includes the Barthel index [43], clinical frailty scale [44], and Fried frailty phenotype defined by weight loss, weakness of hand grip, exhaustion, slowness, and low activity (Table 2) [45]. The cumulative deficit model, on the other hand, is a measure of the accumulation of symptoms, function, comorbidities, clinical laboratory abnormalities, and questionnaire of quality of life, which is represented by the Rockwood frailty index using 93 variables (Table 2) [42]. Although various scales have been used in recent HF studies, the following scales are commonly used: Fried frailty phenotype, Rockwood frailty index, Barthel index, and clinical frailty scale [56]. The Rockwood frailty index has recently been adopted as an evaluation scale for frailty in DAPA-HF [57] and DELIVER trials [58], both of which showed the efficacy of SGLT-2is for HF patients with frailty, and these attracted much attention. However, there is no fixed consensus on the cutoff value for these frailty diagnostic scales.

It should not be forgotten that there is regional variability in the prevalence of frailty. A recent meta-analysis reported that the prevalence of frailty in an Asian population aged over 60 years was 20.5% [59], which was roughly equal to those reported in Latin American and Caribbean populations [60], but higher than in European, North American, and Oceanian populations [61,62,63]. However, due to the lack of a uniform evaluation scale, we need to be cautious to interpret these data of regional differences, suggesting the difficulty of making an unbiased regional comparison and development of a global countermeasure against this issue.

## 3. Heart Failure and Frailty/Sarcopenia

Body weight loss in HF patients was called “cardiac cachexia” especially with a change in body composition [64]. Cachexia is a concept that includes skeletal muscle wasting, anemia, anorexia, and altered immune function, which results in fatigue, impaired quality of life, and an aggregate prognosis [65], and it can occur in patients with a variety of diseases such as HF, chronic obstructive pulmonary disease, renal failure, and cancer. It is different from sarcopenia in terms of its concept, which is not limited to muscle weakness [64]. In HF patients, dyspnea, fatigue, and anorexia can lead to a low nutritional state and reduction in physical activity, which leads to sarcopenia, and further weakening of muscle or physical function. This vicious circle is called the “frailty cycle” [66] (Figure 1).

It is well known that sarcopenia and frailty are strongly associated with a poor prognosis in HF patients. HF patients have a higher prevalence of sarcopenia (by ~20%) compared to healthy subjects of the same age and it is associated with worse clinical outcomes independently [67]. Frailty is prevalent in HF patients, representing 40–80% of overall HF, 30–60% of HF with a reduced ejection fraction (HFrEF), and up to 90% of HF with a preserved ejection fraction (HFpEF) [68,69,70,71,72,73]. In the FRAIL-HF study, HF patients with frailty showed a higher prevalence of depression, worse score of health literacy, few HF medications, and higher risk of mortality and rehospitalization [74]. A recent meta-analysis reported that the presence of frailty in chronic HF is associated with an increased risk of death and hospitalization by approximately 1.5-fold [75]. The reasons why frailty is associated with a worse prognosis are related to HF aggravation by comorbidities such as anemia and renal dysfunction, muscle weakness leading to increased cardiac load [76], difficulty in initiating medications due to organ dysfunction or fall risk by drug-induced hypotension or dehydration, and lower adherence to medication because of cognitive or social frailty.

In addition, in the late 20th century, a subset of older adults was identified as having both obesity and sarcopenia, soon thereafter termed as “sarcopenic obesity”. Sarcopenic obesity is defined by excess adiposity with a loss of muscle mass and/or function [77]. Aging is a systemic process affecting all organ systems and associated with significant alterations in body composition. Typically, fat mass increases with age [78], whereas muscle mass and strength start to decline progressively [79]. While aging is associated with a systemic pro-inflammatory state, oxidative stress, and altered endocrine function leading to the loss of muscles [80], obesity has multiple adverse consequences for skeletal muscle, including inflammation, oxidative stress, and insulin resistance. Along with visceral fat accumulation, loss of skeletal muscle, which is the largest insulin-responsive target tissue, produces insulin resistance. Adding to this, increases in visceral fat may lead to a higher secretion of pro-inflammatory adipokines that further promote insulin resistance as well as potentially direct catabolic effects on muscles [81,82]. The reports on the epidemiology of sarcopenic obesity are limited, but in a 14-year prospective study of the elderly population in the United States, its prevalence was 19–34% in women and 13–27% in men [83]. In the HF population, the prevalence of sarcopenic obesity was reported to be 4.0–18.5% [33,84]. Coexistence of sarcopenic obesity is a predictor of disability and mortality [85,86], and associated with a reduction in cardiorespiratory fitness independent of adiposity [87]. However, the data on its pathophysiology and prognostic impact compared to lean sarcopenia are needed.

## 4. Safety and Efficacy of SGLT-2is for Sarcopenic or Frail Patients

The hypothetical mechanisms mediating the efficacy of SGLT-2is for HF patients are the following: cardio–renal coupling, ketone production, diuretic effect, hematopoietic effect, direct prevention of myocardial remodeling, and suppression of neurohumoral factor [2], and it is considered that each of them have interrelated effects. Since many studies have recently reported the efficacy of SGLT-2is for HF regardless of the history of DM [18,19], their efficacy seems to be not only related to the blood-glucose-lowering effect. Further, the beneficial effect was observed regardless of left ventricular EF [14,19,20,21,88,89,90,91]. Some randomized controlled studies have carried out sub-analysis that focused on frailty (Table 3). In the DELIVER trial, the presence or severity of frailty was assessed for 6258 study patients by their frailty index (FI) at baseline and they were divided into four classes by their FI [92]. The beneficial effect of dapagliflozin on clinical outcome was observed consistently across the FI values, greater improvement in quality of life with treatment occurred in patients with a higher level of frailty, and there were no differences in the proportions of patients who experienced adverse events or discontinued treatment between dapagliflozin and the placebo [58]. Although this study concluded that SGLT-2is may demonstrate efficacy and safety for HF patients even with frailty, there are several limitations in this study. The FI is derived from a cumulative deficit model composed of symptoms, comorbidities, disabilities, tests of muscle weakness, and laboratory data including indices of malnutrition, kidney failure, anemia, and thyroid hormone. In other words, the FI is a comprehensive vulnerability scale and might not be suitable for the evaluation of physical frailty or sarcopenia alone. Similarly, we can point out this weakness of the FI as an assessment scale of frailty in a sub-analysis of DAPA-HF trial, in which the treatment effect by dapagliflozin on the reduction in primary endpoint reduction was greater in patients with a higher degree of frailty defined by the FI [58]. In the DELIVER trial, despite the absence of exclusion criteria related to a low BMI, the average BMI was as high as 32.1 in the “most frail” group. As mentioned in the previous section, there are substantial differences in body composition or mass between Caucasian and Asian populations [47]. While many Caucasian HF patients are deemed to have sarcopenic obesity, most Asian patients show a low BMI. In this regard, patients with widely varying body compositions can be uniformly categorized as “frailty”. FI is a cumulative deficit model for frailty and does not necessarily evaluate physical function or phenotype. Thus, it needs careful consideration when we determine the efficacy and safety of SGLT-2is for the population with “frailty”. In other study, following the sub-analysis of the EMPEROR-Reduced trial, it can be observed that the efficacy of empagliflozin is consistent regardless of BMI, even at <20 kg/m^2^ [93]. However, the clinical evidence of SGLT-2is for HF patients with sarcopenia or physical frailty is limited and needs to be explored in the near future.

A recent study showed SGLT-2is are more efficacious for a primary prevention compared to DPP-4 inhibitors in type 2 DM patients with frailty assessed by FI [96]. This study population included type 2 DM patients over 65 years and patients enrolled in Medicare who initiated treatment with SGLT-2is or DPP-4 inhibitors. SGLT-2is were associated with improved cardiovascular outcomes and all-cause mortality, with the largest absolute benefits among patients with frailty. We should take care in interpreting these data because the FI was used to carry out the frailty assessment and the large number of obese patients included was the same as previous HF studies. Further, genital infections were observed among patients who received SGLT-2i and caused greater harm among more frail patients. Infections can worsen HF and ketoacidosis and are sometimes fatal. Therefore, patients should still be carefully selected for the initiation of SGLT-2is treatment.

SGLT2is have been shown to significantly reduce body weight and fat mass and this effect may be beneficial to improve glycemic control and HF [98]. On the other hand, skeletal muscle mass has also been reported to be significantly reduced [98], although a recent report showed that empagliflozin induced a significant reduction in body weight, body fat mass and water volume, but the skeletal muscle mass did not change significantly in type 2 DM patients aged ≥ 65 years [97]. Thus, significant concerns have been raised about SGLT-2is’ effect on aggravating frailty or sarcopenia. This poses the following question: which patients can safely receive SGLT-2is? Even in DM patients, there is still no unified tool for the assessment of frailty and the guideline recommendations do not address frail older patients [99]. Therefore, it is necessary to consider the metabolic phenotypes of heterogeneous frail patients with DM in order to evaluate the influence of SGLT-2is on these patients. Compared to type Ⅰ muscle fibers, type II fiber is associated with an increase in insulin resistance via lipid storage in muscle tissue [100]. Aging is related to an increase in insulin resistance followed by a loss of muscle fiber; however, frailty is associated with accelerated muscle loss compared with age alone with a prominent reduction in type II (rather than type I) fibers, which may result in an overall reduction in insulin resistance. Thus, it is important to assess the metabolism spectrum by considering the loss of muscle fibers and body adipose/muscle tissue ratio and not only the BMI. Classification into two phenotypes has been proposed: the anorexic malnourished (A..22..2223M) frail phenotype with significant muscle loss and the sarcopenic obese (SO) frail phenotype with increased visceral fat [99]. SGLT-2is could be effective for SO phenotype patients, but their use for AM phenotype patients may exacerbate sarcopenia. Luseogliflozin and canagliflozin have shown minimal reductions in skeletal muscle mass in not-severely overweight patients with type 2 DM [101,102,103]. In opposition, dapagliflozin did not show this effect [104] and another study reported that SGLT-2is improve grip strength [105]. Administration of SGLT-2is for AM phenotype patients may lead to an increase in calorie intake and control of weight loss; however, this effect is dependent on the patient’s insulin secretory capacity and it is necessary to identify target patients.

## 5. Future Direction

As discussed above, the most critical issue is that there are few studies validating the efficacy and safety of SGLT-2is for HF patients with physical frailty (not evaluated by the cumulative deficit model) and/or sarcopenia and there have not yet been unified assessment scales for sarcopenia/frailty for HF. In near future further exploration such as basic research (i.e., experiment using cachexia animal model) is necessary to for further understanding of the pathophysiology of sarcopenia and/or physical frailty and the safety and efficacy of SGLT-2is for patients affected with these conditions. Further, we need to develop assessment tool of sarcopenia and/or physical frailty for HF patients which is useful and readily available in various situations in clinical practice, and reassess the efficacy and safety of SGLT-2is by those indicators in specific populations focused on body size, age, gender, and ethnic differences.

## 6. Conclusions

The efficacy of SGLT-2is for HF patients has been known widely. Beyond the poor definition of frailty of elderly patients suffering from HF, there seems to be an advantage in taking SGLT-2i. However, its long-term safety has not been sufficiently explored and still remains unclear, especially in those with sarcopenia or physical frailty. According to remarkably accelerated aging and increasing prevalence of frailty or sarcopenia in HF population, it is crucial to construct a unified evaluation scale and conduct large-scale clinical trials focusing on the safety and efficacy of SGLT-2is for HF patients with sarcopenia/physical frailty.

## Figures and Tables

**Figure 1 jpm-14-00141-f001:**
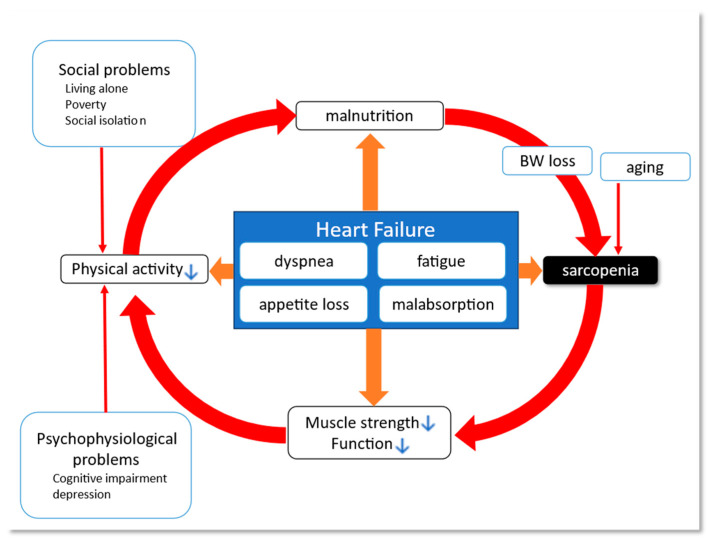
Frailty cycle in HF. Malnutrition can cause body weight loss and muscle loss (sarcopenia), accompanied by deteriorated muscle strength and function, which results in depressed physical activity. As a result, the decrease in oral intake induces further malnutrition. This vicious cycle is often referred to as the frailty cycle. In the setting of HF, the symptoms include dyspnea, fatigue, appetite loss and intestinal malabsorption due to intestinal congestion and malperfusion. These symptoms can accelerate every single step in the frailty cycle. Further, elderly populations are commonly affected by HF and have various problems that further accelerate the frailty cycle. Red arrows indicate the main pathway of the frailty cycle. Orange arrows indicate the aggravation of each component. ↓, decrease. BW, body weight; HF, heart failure.

**Table 1 jpm-14-00141-t001:** The landmark trials that assessed the safety and efficacy of SGLT-2is and their main findings.

Population	SGLT-2is	Trial	Primary Endpoint
T2DM and high risk of CVD	Empagliflozin	EMPA-REG OUTCOME [14]	MACE, HR 0.86 [95%CI, 0.74–0.99]
	Canagliflozin	CANVAS program [15]	MACE, HR 0.86 [95%CI, 0.75–0.97]
	Dapagliflozin	DECLARE-TIIM 58 [16]	The composite of CV death and hospitalization for HF, HR 0.83 [95%CI, 0.73–0.95]
HFrEF	Empagliflozin	EMPEROR-Reduced [18]	The composite of CV death and hospitalization for HF, HR 0.75 [95%CI, 0.65–0.86]
	Dapagliflozin	DAPA-HF [17]	The composite of CV death and hospitalization or urgent intravenous therapy for HF, HR 0.74 [95%CI, 0.65–0.85]
T2DM and HF	Sotagliflozin	SOLOIST-WHF [19]	The composite of CV death and hospitalization or urgent visit for HF, HR 0.67 [95%CI, 0.52–0.85]
HFpEF	Dapagliflozin	DELIVER [20]	The composite of CV death and hospitalization for HF, HR 0.82 [95%CI, 0.73–0.92]
	Empagliflozin	EMPEROR-Preserved [21]	The composite of CV death and hospitalization for HF, HR 0.79 [95%CI, 0.69–0.90]
Acute HF	Empagliflozin	EMPULSE [26]	The composite of all-cause death, worsening HF event, and KCCQ-TSS, stratified win ratio 1.36 [95%CI, 1.09–1.68]

HR, hazard ratio; CI, confidence interval; T2DM, type 2 diabetes mellitus; CVD, cardiovascular disease; MACE, major advanced cardiovascular events (defined as the composite of cardiovascular death, nonfatal myocardial infarction, and nonfatal stroke); HF, heart failure; HFrEF, heart failure with reduced ejection fraction; HFpEF, heart failure with preserved ejection fraction; KCCQ-TSS, The Kansas City Cardiomyopathy Questionnaire Total Symptom Score; EMPA-REG, The Empagliflozin Cardiovascular Outcome Event Trial in Type 2 Diabetes Mellitus Patients–Removing Excess Glucose; CANVAS, Canagliflozin Cardiovascular Assessment Study; DECLARE-TIMI, The Dapagliflozin Effect on Cardiovascular Events–Thrombolysis in Myocardial Infarction; EMPEROR-Reduced, The Empagliflozin Outcome Trial in Patients with Chronic Heart Failure and Reduced Ejection Fraction; DAPA-HF, Dapagliflozin and Prevention of Adverse Outcomes in Heart Failure; DELIVER, Dapagliflozin Evaluation to Improve the Lives of Patients with Preserved Ejection Fraction Heart Failure; SOLIST-WHF, the Effect of Sotagliflozin on Cardiovascular Events in Patients with Type 2 Diabetes Post Worsening Heart Failure; DELIVER, The Dapagliflozin Evaluation to Improve the Lives of Patients with Preserved Ejection Fraction Heart Failure; EMPEROR-Preserved, The Empagliflozin Outcome Trial in Patients with Chronic Heart Failure with Preserved Ejection Fraction; EMPULSE, Empagliflozin in Patients Hospitalized With Acute Heart Failure Who Have Been Stabilized.

**Table 2 jpm-14-00141-t002:** The main tools for the assessment of frailty or sarcopenia.

Frailty or Sarcopenia	Assessment	Measure	Description
Sarcopenia	Muscle mass	Skeletal muscle mass index (SMI) (appendicular skeletal muscle mass/height^2^)	Various cutoffs employed by studies
	Muscle strength	Hand grip	Various cutoffs employed by studies
Sarcopenia/Frailty	Physical function	Gait speed	
	Physical function	Short Physical Performance Battery (SPPB) [40]	A summation of scores on three tests: balance, gait speed and chair stand
	Physical function	Timed-Up and Go test (TUG) [41]	
Frailty	Multidimensional	Rockwood frailty index [42]	Accumulation of symptoms, function, comorbidities, clinical laboratory abnormalities, and impaired quality of life are assessed using 93 variables
	Phenotype model	Barthel index [43]	Score is calculated based on several daily activities (feeding, bathing, grooming, dressing, bowel and bladder control, toilet use capability, transfer from bed to chair and vice-versa, mobility on level surfaces, and capability to climb stairs)
	Medical domain	Clinical frailty scale [44]	A semi-quantitative global judgement
	Medical domain and physical function	Fried frailty phenotype [45]	Weight loss, weakness of hand grip, exhaustion, slowness, and low activity

**Table 3 jpm-14-00141-t003:** Previous SGLT-2i studies that focused on sarcopenia/frailty, BMI or the elderly.

Population	Study	Topics of Interest (Assessment Tool)	Main Findings
HFrEF	DAPA-HF sub-analysis [57]	Frailty (Frailty index)	The efficacy of dapagliflozin for HFrEF patients was consistent across the range of frailty, and the absolute reductions were larger in more frail patients.
	DAPA-HF sub-analysis [94]	BMI	The efficacy of dapagliflozin for HFrEF patients was consistent across the spectrum of BMI.
	EMPEROR-Reduced sub-analysis [93]	BMI	The efficacy of dapagliflozin for HFrEF patients was consistent across the spectrum of BMI, and weight loss was associated with higher all-cause mortality regardless of BMI groups.
HFpEF	DELIVER sub-analysis [58]	Frailty (Frailty index)	The benefit of dapagliflozin for HFpEF patients was consistent across the range of frailty and the improvement of QOL with medication was greater in those with a higher level of frailty.
	DELIVER sub-analysis [95]	BMI	The benefit of dapagliflozin for HFpEF patients was consistent across the spectrum of BMI.
DM	Kutz et al. (2023) [96]	Frailty (Frailty index)	Medicare beneficiaries with type 2 DM showed greater cardiovascular effectiveness associated with SGLT-2is and GLP-1 receptor agonists than DPP-4 inhibitors.
	EMPA-ELDERLY [97]	Elderly (≥65)	Empagliflozin for elderly T2DM reduced body weight without compromising muscle mass or strength.

SGLT-2i, sodium–glucose cotransporter 2 inhibitor; BMI, body mass index; HFrEF, heart failure with reduced ejection fraction; HFpEF, heart failure with preserved ejection fraction; DAPA-HF, Dapagliflozin and Prevention of Adverse Outcomes in Heart Failure; EMPEROR-Reduced, The Empagliflozin Outcome Trial in Patients with Chronic Heart Failure and Reduced Ejection Fraction; DELIVER, Dapagliflozin Evaluation to Improve the Lives of Patients with Preserved Ejection Fraction Heart Failure; EMPA-ELDERLY, Empagliflozin in Elderly T2DM Patients; DM, diabetes mellitus; GLP-1, glucagon-like peptide 1; DPP-4, dipeptidyl-peptidase 4.

## Data Availability

Not applicable.

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
