# Peer review of "The Safety and Efficacy of Sodium-Glucose Cotransporter-2 Inhibitors for Patients with Sarcopenia or Frailty: Double Edged Sword?"

_jpm, 2024, doi:10.3390/jpm14020141_

Round 1
Reviewer 1 Report
Comments and Suggestions for Authors
I read with great interest the work of Ayami Naito and colleagues which highlights the importance of a thoughtful assessment of the frailty of elderly patients in order to understand the actual safety and effectiveness of SGLT-2i. As underlined by the authors, many studies have highlighted a real advantage in taking this category of drugs even in the frail elderly population, highlighting however that in these studies the definition of frailty often stops at sarcopenia alone. The concept of frailty in geriatric medicine is widely and precisely described and is not limited to sarcopenia alone.
Nonetheless, although the method of defining frailty in the large clinical trials that observed the effects of SGLT-2i is perhaps not very precise, there is still an advantage in their intake even in the categories of patients selected as frail. The difference in the body composition of Causal patients compared to Asian ones, pointed out by the authors as a possible confounding factor, could be easily overcome with a large population study in the frail and non-fragile Asian population, to evaluate the actual safety and effectiveness also in that context .
I would therefore explicitly write in the conclusions that, beyond the poor definition of "frailty" of elderly patients suffering from heart failure, there seems to be an advantage in taking SGLT-2i. However, studies in specific populations are recommended, to observe the efficacy and safety specifically (even in the female population compared to the male one, or in African ethnic groups it would be interesting to evaluate also based on specific genetic differences).
Valuable work.
Minor points:
Please, check some typos in the text (e.g. Line 31-32: “mediating” (attention to clearly formatting the text).
Comments on the Quality of English Language
No English language editing needed
Author Response
We would like to thank the Editors and the Reviewers for their detailed and thoughtful comments and suggestions. We have carefully considered each point and this has guided our revision of the previous manuscript as detailed below.
In the following response letter, original Editor/Reviewer comments are listed in bold, our reply is in regular font, and verbatim textual changes to the manuscript are in italics.
Reviewer 1:
I read with great interest the work of Ayami Naito and colleagues which highlights the importance of a thoughtful assessment of the frailty of elderly patients in order to understand the actual safety and effectiveness of SGLT-2i. As underlined by the authors, many studies have highlighted a real advantage in taking this category of drugs even in the frail elderly population, highlighting however that in these studies the definition of frailty often stops at sarcopenia alone. The concept of frailty in geriatric medicine is widely and precisely described and is not limited to sarcopenia alone.
We sincerely thank the Reviewer for his/her positive comment on our manuscript and the excellent suggestion below.
Nonetheless, although the method of defining frailty in the large clinical trials that observed the effects of SGLT-2i is perhaps not very precise, there is still an advantage in their intake even in the categories of patients selected as frail. The difference in the body composition of Causal patients compared to Asian ones, pointed out by the authors as a possible confounding factor, could be easily overcome with a large population study in the frail and non-fragile Asian population, to evaluate the actual safety and effectiveness also in that context.
I would therefore explicitly write in the conclusions that, beyond the poor definition of "frailty" of elderly patients suffering from heart failure, there seems to be an advantage in taking SGLT-2i. However, studies in specific populations are recommended, to observe the efficacy and safety specifically (even in the female population compared to the male one, or in African ethnic groups it would be interesting to evaluate also based on specific genetic differences).
Valuable work.
We appreciate the Reviewer’s insightful comments. Indeed, the differences in body composition between Asians and Caucasians can be compensated by comparisons within the same ethnic group including a large number of patients. However, as the working groups from Europe and Asia proposed quite different criteria for the diagnosis of sarcopenia, it is remain uncertain whether different ethnic groups should be assessed with the same scale or different one when determining the efficacy and safety of certain medication like SGLT-2i.
Certainly, SGLT-2is are effective in various patients with HF, however its efficacy has not been fully explored especially in leaner patients (BMI<18.5) even in sub-analysis by body weight of DAPA-HF trial (Eur J Heart Fail 2021; 10: 1662-72). Further large studies on these issues are expected in the future for personalized medicine. Reflecting reviewer’s comment, we have now added the description shown below in the Future Perspective.
Page 7:
Further, we need to develop assessment tool of sarcopenia and/or physical frailty for HF patients which is useful and readily available in various situations in clinical practice, and reassess the efficacy and safety of SGLT-2is by those indicators in specific populations focused on body composition, age, gender, and racial differences.
Further, we agree to the conclusions raised by the Reviewer. We added the description shown below in the Conclusions.
Page 7, lines
Beyond the poor definition of frailty of elderly patients suffering from HF, there seems to be an advantage in taking SGLT-2i.
Minor points:
Please, check some typos in the text (e.g. Line 31-32: “mediating” (attention to clearly formatting the text).
We appreciate the reviewer’s indication. We have corrected the typos.
Reviewer 2 Report
Comments and Suggestions for Authors
The paper is interesting, but adequate editing is required.
As an example both abbreviations "SGLT-2is" (see line 11) and "SGLT-2i" (see line 28) are used for the same term SGLT-2 inhibitors: please correct all abbreviations in a unique modality throughout the paper.
Lines 31-32: "The various mechanisms mediating its beneficial effect have been proposed. [2]." Not clear. Please report and explain the mechanisms.
All along the paper, references are often reported after the point or the comma of the pertinent phrase as in line 32: ..." have been proposed. [2]".... or in line 47, 48, 61, 65, 70, 72, 79, 91, 95, 97, 100, 102, 128, 164, 166, 187, 197, 208, 209, 212, 215 and so on. Please correct all.
Please avoid unnecessary bold character: see lines 52, 64, 88,106,109.
Line 71: use "ethnic" instead of "racial" term.
In Tabel 1 use Capital characters for the initials of the words (Sarcopenia, Frailty). Correct position of the word Multidimension...
Line 145: PMID: 37586848: Report in the References the complete reference with the pertinent number: Talha KM, Pandey A, Fudim M, Butler J, Anker SD, Khan MS. Frailty and heart failure: State- of-the-art review. J Cachexia Sarcopenia Muscle. 2023 Oct;14(5): 1959-1972. doi: 10.1002/ jcsm.13306.
Line 153: report the right reference number at the place of "(ref)".
Lines 154-155: ...."drug-induced hypotension or dehidration"...
Line 159: "quote" instead of "quate" ?
Lines 197-198: better to put reference number at the end of the phrase.: "A recent study showed..."
Line 223: "ratio" instead of "ration" ?
Line 240: "there is" or "there are" ?
Line 244: ..."is necessary to for further understanding..." please correct.
Line246: Please rephrase the text.
Comments on the Quality of English LanguageModerate editing is required.
Author Response
We would like to thank the Editors and the Reviewers for their detailed and thoughtful comments and suggestions. We have carefully considered each point and this has guided our revision of the previous manuscript as detailed below.
In the following response letter, original Editor/Reviewer comments are listed in bold, our reply is in regular font, and verbatim textual changes to the manuscript are in italics.
Reviewer 2:
The paper is interesting, but adequate editing is required.
We sincerely thank the Reviewer for his/her positive comment on our manuscript.
As an example both abbreviations "SGLT-2is" (see line 11) and "SGLT-2i" (see line 28) are used for the same term SGLT-2 inhibitors: please correct all abbreviations in a unique modality throughout the paper.
We have corrected spelling inconsistencies and unified its notation to “SGLT-2is”.
Lines 31-32: "The various mechanisms mediating its beneficial effect have been proposed. [2]." Not clear. Please report and explain the mechanisms.
We described the “various mechanisms” induced by SGLT-2 inhibitors in detail.
All along the paper, references are often reported after the point or the comma of the pertinent phrase as in line 32: ..." have been proposed. [2]".... or in line 47, 48, 61, 65, 70, 72, 79, 91, 95, 97, 100, 102, 128, 164, 166, 187, 197, 208, 209, 212, 215 and so on. Please correct all.
We have corrected all citations prior to period or commas according to the Reviewer’s comments.
Please avoid unnecessary bold character: see lines 52, 64, 88,106,109.
We have unbolded all according to the Reviewer’s comments.
Line 71: use "ethnic" instead of "racial" term.
We have corrected it.
In Tabel 1 use Capital characters for the initials of the words (Sarcopenia, Frailty). Correct position of the word Multidimension...
We have corrected them.
Line 145: PMID: 37586848: Report in the References the complete reference with the pertinent number: Talha KM, Pandey A, Fudim M, Butler J, Anker SD, Khan MS. Frailty and heart failure: State- of-the-art review. J Cachexia Sarcopenia Muscle. 2023 Oct;14(5): 1959-1972. doi: 10.1002/ jcsm.13306.
We amended it according to the Reviewer’s comments.
Line 153: report the right reference number at the place of "(ref)".
We added the appropriate reference.
Lines 154-155: ...."drug-induced hypotension or dehidration"...
Reflecting reviewer’s comment, we have now added the words shown below.
Page 5:
…difficulty in initiating medications due to organ dysfunction or fall risk by drug-induced hypotension or dehydration, and lower adherence to medication because of cognitive or social frailty.
Line 159: "quote" instead of "quate" ?
We have corrected it.
Lines 197-198: better to put reference number at the end of the phrase.: "A recent study showed..."
We amended it according to the Reviewer’s comments.
Line 223: "ratio" instead of "ration" ?
We amended it according to the Reviewer’s comments.
Line 240: "there is" or "there are" ?
“there are” is correct. We amended it according to the Reviewer’s comments.
Line 244: ..."is necessary to for further understanding..." please correct.
We apologize for the typo. “to” was removed.
Line246: Please rephrase the text.
We amended it according to the Reviewer’s comments.
Reviewer 3 Report
Comments and Suggestions for Authors
The manuscript entitled The safety and efficacy of sodium-glucose cotransporter-inhibitors for patients with sarcopenia or frailty: Double edged sword? is a review about the treatment with sodium-glucose cotransporter-inhibitors in patients with sarcopenia or frailty. In this review, the authors summarized the methodology of frailty assessment and discussed the efficacy and safety of SGLT-2i for HF patients with sarcopenia or frailty.
The manuscript is well written and is important for clinical practice.
This is an interesting article, which underscore the safety (more than the efficacy) of SGLT-2i.
Page 4 line 145: Please replace PMID: 37586848 by the appropriate reference.
Please verify the manuscript because there are some editing errors (For e.g. DELIVER trial not DELIVR trial, etc).
Please extend table 2 with all published studies with SGLT-2i (with empagliflozin, canagliflozin). There are only studies with dapagliflozin published until now, in patients with heart failure (with or without DM)?
It would be interesting to add the outcomes on mid or long term (if exist) in these patients with sarcopenia and/or frailty and SGLT-2i.
Otherwise it is a good article with some clinical implications.
Author Response
We would like to thank the Editors and the Reviewers for their detailed and thoughtful comments and suggestions. We have carefully considered each point and this has guided our revision of the previous manuscript as detailed below.
In the following response letter, original Editor/Reviewer comments are listed in bold, our reply is in regular font, and verbatim textual changes to the manuscript are in italics.
Reviewer 3:
The manuscript entitled The safety and efficacy of sodium-glucose cotransporter-inhibitors for patients with sarcopenia or frailty: Double edged sword? is a review about the treatment with sodium-glucose cotransporter-inhibitors in patients with sarcopenia or frailty. In this review, the authors summarized the methodology of frailty assessment and discussed the efficacy and safety of SGLT-2i for HF patients with sarcopenia or frailty.
The manuscript is well written and is important for clinical practice.
This is an interesting article, which underscore the safety (more than the efficacy) of SGLT-2i.
We sincerely thank the review for this positive comment on our manuscript and the excellent suggesting below.
Page 4 line 145: Please replace PMID: 37586848 by the appropriate reference.
We have corrected it.
Please verify the manuscript because there are some editing errors (For e.g. DELIVER trial not DELIVR trial, etc).
We have corrected the typos including the one suggested by the Reviewer.
Please extend table 2 with all published studies with SGLT-2i (with empagliflozin, canagliflozin). There are only studies with dapagliflozin published until now, in patients with heart failure (with or without DM)?
Table 2 (NEW Table 3) shows the post-hoc analysis of the landmark trials which sought to determine the efficacy and safety of SGLT-2is for patients with frailty. We created NEW Table 1 which summarized the landmark trials demonstrating SGLT-2is’ effect for patients with DM and/or HF.
It would be interesting to add the outcomes on mid or long term (if exist) in these patients with sarcopenia and/or frailty and SGLT-2i.
Otherwise it is a good article with some clinical implications.
We appreciate reviewer’s insightful indication and positive comment on our manuscript. To our knowledge, the studies which are shown in Table 2 (NEW Table 3) are all which focused on patients with sarcopenia and/or frailty and SGLT-2i.
Round 2
Reviewer 3 Report
Comments and Suggestions for Authors
Thank you for responding me to my comments.